# LOCAL STABILITY OF ADVERSARIAL EXAMPLES

**I-Jeng Wang, Michael Pekala, Evan Fuller**[*]
Johns Hopkins University Applied Physics Laboratory
Laurel, MD 21771, USA
`firstname.lastname@jhuapl.edu`

## ABSTRACT

Neural networks' lack of stability to "small" perturbations of the input signal is a topic of substantial interest. Recent work has explored a number of properties of these so-called adversarial examples (AE) in an attempt to further understand this phenomenon. The present work continues in this spirit and provides an explicit characterization of stability for AE derived from very small perturbations. We also suggest future directions for how this characterization might be used in practice to mitigate the impact of these AE.

## 1 INTRODUCTION

An outstanding open question for the study of adversarial examples is: Do there exist intrinsic properties of AEs that are invariant across a wide range of generation methods? Also, what minimal assumptions on the adversarial perturbations are needed in order to identify unique properties of the resulting adversarial examples? A number of properties of adversarial examples have been proposed and explored in the literature. From the onset it was hypothesized that AE are unusual events in some sense, occupying relatively small regions of input space which explains why they are rarely observed by happenstance Szegedy et al. (2013). The role of linearity and consequences of using $\ell_\infty$ norms to measure perturbations (vs. other $p$-norms) has also been proposed as a way to understand AE Goodfellow et al. (2014). More recent work suggests that AE are not scattered arbitrarily, but rather occupy non-trivial contiguous regions in the vicinity of clean examples Tramèr et al. (2017). Topological properties that are less local in nature, such as the path connectedness of these AE regions, have also been considered Fawzi et al. (2017).

While the notion of what constitutes a "small" perturbation may be application-dependent, one can argue that AE which arise from the smallest perturbations are the most conceptually interesting (otherwise, the discussion of AE becomes confounded with related topics, such as outlier detection). In this case, local properties of the underlying network are relevant. Another interesting empirical observation is that AE lying close to a classification boundary tend to be unstable; there are many small perturbations that cause the classifier to change its decision. This dichotomy between behavior of the loss function near AE versus near clean examples motivated some authors to propose a Monte Carlo Sampling scheme that can mitigate subtle AE while preserving proper classification on clean examples Cao & Gong (2017). While other interesting pre-processing based AE defenses have also been proposed (e.g. Guo et al. (2017)), it is this notion of stability that we explore in section 2.

## 2 LOCAL STABILITY OF ADVERSARIAL EXAMPLES

In the following, we will use $h \colon \mathbb{R}^d \to \mathcal{Y}$ to denote a deep network model, where $\mathcal{Y}$ is finite set of possible class labels. We will use $\boldsymbol{x} \in \mathbb{R}^d$ to represent a generic input to $h$. Hence $h(\boldsymbol{x})$ is the label of $\boldsymbol{x}$ based on model $h$. We start with a general definition of adversarial examples:

**Definition 1.** *Given a (clean) example $\boldsymbol{x} \in \mathbb{R}^d$, an example $\boldsymbol{x}' \in \mathbb{R}^d$ is called an $\epsilon$-adversarial example (AE) of $\boldsymbol{x}$ for model $h$ if $\boldsymbol{x}' \in B(\boldsymbol{x}, \epsilon)$ and $h(\boldsymbol{x}') \neq h(\boldsymbol{x})$, where $B(\boldsymbol{x}, \epsilon)$ denotes the closed $\epsilon$-ball[1] around $\boldsymbol{x}$.*

---

[*]Visiting scientist
[1]Here we restrict our discussion to the $L_2$ norm.

The above definition is consistent with the standard notion of the AE studied in the literature and does not depend on how the AE is generated. To permit a non-trivial analysis of adversarial examples, we argue that we need additional assumption on the "clean" example $x$. We introduce the following local stability property of $h$:

**Definition 2.** *An example $x \in \mathbb{R}^d$ is called an $\epsilon$-stable ($\epsilon > 0$) example for model $h$ if the set $S_h(x, \epsilon)$ defined below has (Lebesgue) measure zero:*

$$S_h(x, \epsilon) \triangleq \{x' : x' \text{ is an } \epsilon\text{-AE of } x\}.$$

Equivalently, we can say $x$ is an $\epsilon$-stable example if a (uniformly) random sample from $B(x, \epsilon)$ has *probability zero* of having a different label from $x$. In contrast to the assumption typically made in the AE literature that $h(x)$ is the "true" label, We argue that it is important to charcterize the local stability of the classifer at $x$: From any example that is not stable, random perturbations are likely to result in different labels; hence the robustness to AE may be of secondary concern.

Without further assumptions about an example $x$ and its associated AE $x'$, one can establish the following result that is a direct consequence of definitions 1–2:

**Lemma 1.** *Suppose $x$ is $\epsilon$-stable and let $x'$ be an $\tau$-AE of $x$. If $\tau < 2\epsilon$, then $x'$ cannot be $\epsilon$-stable.*

*Proof.* Because $x$ and $x'$ are within $2\epsilon$ of each other, the intersection of the $\epsilon$-balls around them has strictly positive measure, that is,

$$\mu\left(B(x, \epsilon) \cap B(x', \epsilon)\right) > 0,$$

where $x$ is an $\epsilon$-stable example and $x'$ its associated $\tau$-AE, and $\mu$ is the Lebesgue measure in $\mathbb{R}^d$. Since $x$ is $\epsilon$-stable, vectors in the intersection have different label from $x'$ with probability one. Hence $x'$ is not $\epsilon$-stable. $\qquad\square$

**Remark:** Lemma 1 basically states that AE lying sufficiently close to $\epsilon$-stable clean examples cannot themselves be $\epsilon$-stable (they must have a smaller stability region). This is compatible with the intuition behind the sampling-based AE defense of Cao & Gong (2017); that is, uniformly sampling nearby test points is unlikely to disrupt clean, stable examples while possibly disrupting AE. In fact, Lemma 1 provides further insight into the effectiveness of this defense heuristic: it is effective when an AE lies within the interior of the stability region of the clean example; and its performance degrades as the AE moves outside the stability region. Experiment results presented in the next section illustrate this property.

## 3 Preliminary Experiments and Discussion

Lemma 1 also suggests that it may be possible to detect AE using $\epsilon$-stability as a criterion[2]. This task proves to be challenging because (1) directly measuring $\epsilon$-stability is difficult in high-dimensional space where naive sampling strategies do not provide sufficient statistical significance, and (2) the stability level $\epsilon$ may vary significantly from one example to the next, making it difficult to design robust statistics without making use of additional properties of the network $h$.

Despite these challenges, we consider preliminary experiments that explore the notion of relative $\epsilon$-stability for clean and adversarial example pairs. Of course, this requires a mechanism for approximating $\epsilon$. We observe that the minimum distance to a classification boundary taken from a small collection of randomly sampled directions emanating from a point provides a (loose) upper bound on $\epsilon$-stability for that point. In other words, if a $d$-AE is found by such a simple sampling scheme, it is reasonable to conclude that $\epsilon < d$. For our experiments, the $\epsilon$-stability of a point is estimated by finding the nearest classification boundary taken from a set of 100 randomly sampled directions and dividing this distance by a constant factor (here, five was used). Obviously this estimate is not precise and is neither guaranteed to be an upper nor a lower bound; this is meant simply as a starting point for our investigations.

---

[2]Note that, while a number of methods for detecting AE have been proposed (e.g. see Carlini & Wagner (2017) for a nice summary and related challenges); what differentiates our agenda here is a attempt to base the detection scheme upon explicit characteristics of AE (e.g. rather than a black-box model).

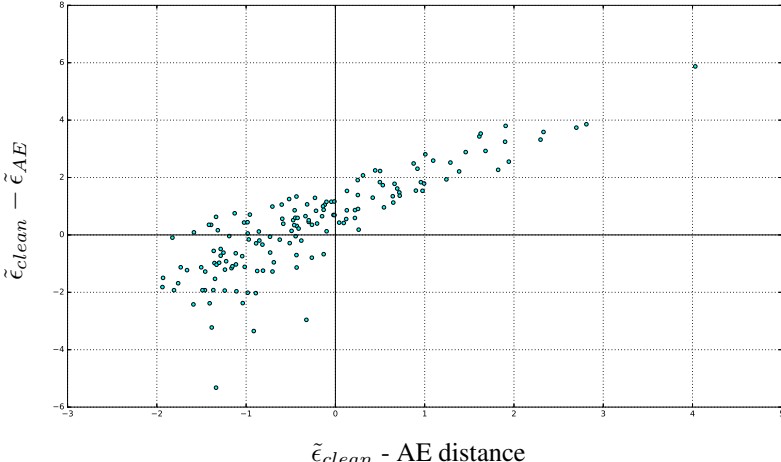

$\tilde{\epsilon}_{clean}$ - AE distance

Figure 1: Relative $\epsilon$-stability for clean and adversarial examples (where $\tilde{\epsilon}_{clean}$ and $\tilde{\epsilon}_{AE}$ are the estimated stability levels for clean example and AE, respectively; and AE distance is the distance between an AE and its associatd clean example).

In the following, we focus our attention on cases where an AE lies within the $\epsilon$-stable region of the corresponding clean example. While lemma 1 applies to a broader class of AE (up to distance $2\epsilon$), those within the $\epsilon$-stability region will have a more substantial overlap in their stability regions, even with the error induced by our approximation of $\epsilon$.

Figure 1 shows an analysis of $\epsilon$-stability regions for pairs of clean and adversarial examples (generated using the iterative fast gradient method) in the context of the CIFAR-10 classification problem. Each point represents a clean/adversarial example pair. The $x$-coordinate relates the adversarial perturbation magnitude to the (estimated) stability of the corresponding clean example, $\tilde{\epsilon}_{clean}$; points in the right half-plane correspond to AE lying within the $\tilde{\epsilon}_{clean}$-stable region. The $y$-coordinate of each point gives the difference between the clean and adversarial stability region estimates, i.e. $\tilde{\epsilon}_{clean} - \tilde{\epsilon}_{AE}$; points in the upper half plane therefore correspond to instances where the clean example is more locally stable than its adversarial counterpart. This figure shows the key point that, when an AE lies within the $\tilde{\epsilon}_{clean}$-stable region, its stability estimates is indeed less than $\tilde{\epsilon}_{clean}$, in accordance with lemma 1. Similar results were observed for the (non-iterative) fast gradient method and over a variety of perturbation constraints; space limitations preclude presentation of these figures.

While this work is quite preliminary, it suggests a possible approach for characterizing AE by starting with minimal assumptions and building outwards (e.g. by including additional assumptions and knowledge about the loss function, AE generation method, etc.) as required. We point out several future research directions:

- The local nature of lemma 1 suggests that linear analysis methods taking into the local gradient structure of the model (such as the GAAS method of Tramèr et al. (2017)) might provide complementary information; we conducted preliminary experiments along these lines that cannot be included due to space limitations;

- While it may be challenging to estimate the stability of a clean example (we may not have access to it if the example encountered were an AE), it may be possible to construct a model with a desired $\epsilon$-stability by incorporating a notion of robustness into the loss function during training (as explored in Cao & Gong (2017)). Note that one can then establish a theoretical guarantee of robustness to adversarial perturbation of size $\tau$ using the uniform sampling defense scheme if $\tau$ is sufficiently less than $2\epsilon$.

- We plan to expand this study to include other datasets, classifiers, and attacks, which will allow for broader conclusions and local stability patterns.

- We will investigate possible extensions of lemma 1 by relaxinig the stablity condition on the clean examples (e.g., small positive probability vs. zero probability) and/or incorporate additional knowledge of the classifier $h$.

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
