# OpenReview forum: "Local Stability of Adversarial Examples"
_ICLR.cc/2018/Workshop — Reject_

### Official Review · AnonReviewer1 · 2018-03-10
**Good formalism for a start; need more experimental evidence**

**Rating:** 3
**Confidence:** 3

**Review:**

The paper provides a characterization of stability for adversarial examples.
Clarity
--------
1) It is not clear what the goal of the paper is? Is it just limited to formalizing the idea of sampling based AE defense (such as  Cao & Gong (2017))? Or does the work propose something more?

2) The x-axis of Figure 1, says distance and we still find points with negative distance. Please clairfy

Quality
---------
It is a good idea to check how far does the notion local stability differ between clean vs adversarial samples. However, the results presented here seems to be very limited and preliminary with a sample size of just around 100 points. It will be great to have more experimental coverage.

Originality
--------------
Formalization of local stability is important, however, neither the definitions nor Lemma 1 seem to be non-trivial to obtain.

Novelty
----------
Other than formalizing local stability the paper does not find anything drastically 'new' about adversarial examples.


Significance
---------------
The experiments lack coverage, and hence the significance of the work is limited.

---

### Official Review · AnonReviewer2 · 2018-03-10
**Simple (almost trivial) idea, somewhat confusing paper, deeply flawed experiments.**

**Rating:** 3
**Confidence:** 4

**Review:**

The paper defines, for a given trained model h, eps-stability for a clean example x if a random sample x' from a ball of radius eps around x has h(x) = h(x') with probability 1. While the authors couch this in terms of measure theory, I believe that even the tiniest amount of smoothness (which is always present in practice) lets us talk about this differently and more simply: an example is eps-stable if it has at least distance eps from any classification boundary.

The paper has a single lemma, which is that if we have an example that's x eps-stable, and an adversarial example (AE) x' at distance tau < 2*eps from from x, then x' cannot be eps-stable. Thinking in terms of smooth boundaries, this is true but fairy trivial: x has distance >= eps from the classification boundary, x' is on the other side but at distance < 2*eps from x, so x' is at distance < eps from the boundary. Fine.

So what are the implications? The paper is a bit confusing on this point, but I believe the authors want to explore the hypothesis that clean examples x have higher stability than associated AE's x': in other words that a clean example is farther from the classification boundary than an associated AE. If true, this possibly gives a means to detect AE's.

The authors attempt to design an experiment to measure this, but I view the experiment as deeply flawed. The authors *estimate* eps by choosing 100 random directions, finding the distance to the classification boundary, taking the minimum over the 100 directions, and then dividing by 5. They authors correctly point out that this is neither an upper nor a lower bound; they do not mention that the 100 and *especially* the 5 are completely unmotivated fudge factors, and that modifying that 5 may easily change the results of the experiment.

The authors say they want to focus on "cases where an AE lies within the eps-stable region of the corresponding clean example." But this makes no sense: according to lemma 1, AE's within the eps-stable region of x occur w.p. 0, and under a smoothness assumption, there are none of them. Surely the authors are not suggesting that their method of finding AE's is finding measure zero sets? Based on this argument, there are no AE's within the eps-stable region. What the authors actually find is AEs within the ~eps-stable region, where ~eps is the *estimated* stability using the hack described in the previous paragraph. But, continuing the argument, what this is showing is that for any example where the authors find an AE within ~eps, then ~eps is an overestimate. (Put differently, if we *have* an AE x' for x, under very weak smoothness assumptions we can upper bound eps by using the distance to the boundary along the ray from x to x', rather than a random direction.) The authors want to show that for points where the AE is within the eps-stable region, the stability of the AE is lower, but since they're using ~eps which is a (possibly) gross overestimate of eps for these points, I'm not sure what the experiment shows.

I think a better experiment would be to use an upper bound on eps which is the distance to the classifier boundary in the direction of the AE, and then explore whether the distance to the AE < 2 eps. This is still far from perfect, because eps may in fact be quite a bit smaller, but at least the experiment would tell us *something*: if we found that the AE's were (or could be generated to be) often at distance > 2 eps, that this relative stability approach to correcting wouldn't work.

Although the paper is short and simple, it still manages to be confusing. There are many sentences that I can't quite figure out what they're trying to say.

---

### Official Review · AnonReviewer3 · 2018-03-11
**Somewhat trivial results**

**Rating:** 3
**Confidence:** 3

**Review:**

Lemma 1 is almost trivial from the definitions, and I found the directions and claims are not that interesting.
While I agree that the local stability is an interesting property to be examined, it is not interesting if the original prediction is wrong. For example, a classifier that predicts only one label every time is infinity-stable, but not interesting at all since the classifier's prediction is useless. Therefore I would encourage the authors to include the notion of "correct" label in the definition, instead of simply ignoring them.

I also think the vast amount of adversarial example research is focusing on how to efficiently find a such an example, or computing the stability bound, etc. I believe that the theoritical foundation of classifier's stability and its behavior around the decision boundary are heavily investigated in the large margin based classifier, such as SVM, literature. I would suggest to outline the main difference between "margin" and "stability". Also I think it will be interesting to investigate the effect of loss function to the stability region, by training NN using a margin based loss (like hinge loss) instead of logistic loss (or softmax),

---

### Decision · Program_Chairs · 2018-03-20
**ICLR 2018 Workshop Acceptance Decision**

**Decision:**

Reject

**Comment:**

Based on the reviews, this paper has not been accepted for presentation at the ICLR workshop. However, the conversation and updates can continue to appear here on OpenReview.